# Ag Catalysts Supported on CeO_2_, MnO_2_ and CeMnO_x_ Mixed Oxides for Selective Catalytic Reduction of NO by C_3_H_6_

**DOI:** 10.3390/nano13050873

**Published:** 2023-02-26

**Authors:** Eleonora La Greca, Tamara S. Kharlamova, Maria V. Grabchenko, Luca Consentino, Daria Yu Savenko, Giuseppe Pantaleo, Lidiya S. Kibis, Olga A. Stonkus, Olga V. Vodyankina, Leonarda Francesca Liotta

**Affiliations:** 1Institute for the Study of Nanostructured Materials (ISMN), (Italian) National Research Council (CNR), Via Ugo La Malfa 153, 90146 Palermo, Italy; 2Laboratory of Catalytic Research, Tomsk State University, Lenin Ave. 36, 634050 Tomsk, Russia; 3Boreskov Institute of Catalysis SB RAS, Lavrentiev Ave. 5, 630090 Novosibirsk, Russia

**Keywords:** Ag^+^, CeMnO_x_, C_3_H_6_-SCR of NO, oxide microstructure, HRTEM, Raman, XPS

## Abstract

In the present study CeO_2_, MnO_2_ and CeMnO_x_ mixed oxide (with molar ratio Ce/Mn = 1) were prepared by sol-gel method using citric acid as a chelating agent and calcined at 500 °C. The silver catalysts (1 wt.% Ag) over the obtained supports were synthesized by the incipient wetness impregnation method with [Ag(NH_3_)_2_]NO_3_ aqueous solution. The selective catalytic reduction of NO by C_3_H_6_ was investigated in a fixed-bed quartz reactor using a reaction mixture composed of 1000 ppm NO, 3600 ppm C_3_H_6_, 10 vol.% O_2_, 2.9 vol.% H_2_ and He as a balance gas, at WHSV of 25,000 mL g^−1^ h^−1^.The physical-chemical properties of the as-prepared catalysts were studied by several characterization techniques, such as X-ray fluorescence analysis, nitrogen adsorption/desorption, X-ray analysis, Raman spectroscopy, transmission electron microscopy with analysis of the surface composition by X-ray energy dispersive spectroscopy and X-ray photo-electron spectroscopy. Silver oxidation state and its distribution on the catalysts surface as well as the support microstructure are the main factors determining the low temperature activity in NO selective catalytic reduction. The most active Ag/CeMnO_x_ catalyst (NO conversion at 300 °C is 44% and N_2_ selectivity is ~90%) is characterized by the presence of the fluorite-type phase with high dispersion and distortion. The characteristic “patchwork” domain microstructure of the mixed oxide along with the presence of dispersed Ag^+^/Ag_n_^δ+^ species improve the low-temperature catalyst of NO reduction by C_3_H_6_ performance compared to Ag/CeO_2_ and Ag/MnO_x_ systems.

## 1. Introduction

Currently, internal combustion engines (ICEs), including diesel engines, are the most widely used due to their high efficiency and reliability [1]. However, the main disadvantage of the ICEs is the emission of exhaust gases that pose a serious threat to both the environment (increasing ozone concentration in the atmosphere and producing acid rains) and the human health as they are rich in particulates and nitrogen oxides (NO, NO_2_ and N_2_O) [2].

In recent years, several methods have been applied to reduce NO_x_ emissions. For this purpose, selective catalytic reduction (SCR) with hydrocarbons or alcohols (HC- or HCO-SCR, respectively) has proved to be interesting for their high efficiency and low cost [3,4,5]. The main advantage of such reaction is the use of gas mixtures with similar composition as the exhaust fumes; in this way, the HC and NO_x_ can be simultaneously abated without feeding additional reducing agents [6]. Furthermore, this process can be a useful alternative to commercial processes utilizing NH_3_ or urea as reducing agents [7], which are the dominant technologies for NO_x_ removing from mobile (vehicles and marine engines) and stationary sources [8]. Nevertheless, the toxicity of concentrated ammonia, the fact that urea must be temporarily stored on board, thus requiring additional infrastructures for supply and use (additional urea tank to be filled periodically), the costs of ammonia plants and ammonia slip, which can produce additional pollution [5,9], constitute the main obstacles that prevent the large use of such systems.

Numerous catalysts such as zeolites, noble metals and metal oxides have been studied for NO_x_ HC-SCR. The limited use of zeolite-based catalysts is due to hydrothermal deactivation and high-temperature activity limiting their real applications [10]. From the pioneer study by Miyadera [11], based on the performance efficiency of Ag/Al_2_O_3_, which showed high NO conversion in HC-SCR with various light hydrocarbons, the silver-based catalysts are extensively studied as promising candidates for practical use DeNO_x_ systems, because they exhibit a high efficiency comparable to that of commercial catalysts applied in NH_3_-SCR, especially at temperatures above 300 °C; moreover, they have a moderate resistance to water and SO_2_ [12,13,14]. An advantage linked to the preferable use of Ag compared to Pt group metals is a lower oxidation activity of HC/HCO that limits the oxidation of hydrocarbons or oxygenates in simultaneous total combustion during the NO_x_ SCR. Previous studies showed that the catalytic properties of silver-alumina catalysts were related to the Ag loading on the support [15,16]. For low loadings, the Ag exists mainly in the form of Ag^+^ or Ag^δ+^, while the catalysts with higher Ag content usually contain more Ag^0^ nanoparticles. Another important aspect is the influence of Ag loading on the catalytic activity. In fact, isolated silver cations (Ag^+^) and oxidized silver clusters (Ag_n_^δ+^) are proposed to be the active species in the NO-SCR reaction, while metallic silver clusters (Ag^0^) are responsible for the nonselective oxidation of hydrocarbons [16]. However, the practical application of Ag-based catalysts is limited by the low activity at temperatures in the range of 150–300 °C.

Among various supports used for NO_x_ SCR, CeO_2_ and manganese oxides have attracted wide attention as they feature excellent low-temperature activity [17,18,19]. The high efficiency of CeO_2_-based catalysts is due to the excellent Lewis surface acidity, redox properties and high oxygen storage capacity [20,21]. The Mn-based oxides are promising for application at low temperatures due to high NO conversion and good N_2_ selectivity in the NO SCR by NH_3_ [22,23]. Moreover, the Mn-WO_3_/TiO_2_ catalysts represent a valid alternative in the NH_3_-SCR of NO to the typical V_2_O_5_-WO_3_/TiO_2_ commercial systems [24]. According to the available literature data, the Ag catalysts supported on Ce and Ce-based mixed oxides (Ce-Mn, Ce-Zr, Ce-Ti, etc.) are appealing as catalytic materials for NH_3_ NO_x_ SCR in exhausts emitted by diesel engines of vehicles and ships in compliance with the EURO VI and IMO 2020 regulations [14,18,25,26]. However, such catalytic systems remain poorly understood for CH-SCR. Recently, Ag/CeZr catalysts have been shown to be very promising for NO_x_ HC-SCR [27], which increases interest in considering other Ce-based mixed oxides for NO_x_ HC-SCR.

This work is focused on the synthesis and detailed characterization of powder catalysts based on silver as an active phase supported on Ce, Mn and Ce-Mn reducible oxides and study of their activity in selective catalytic NO_x_ reduction with propylene. To synthesize the oxide supports, the citrate sol-gel method was chosen, while an impregnation of the prepared supports with [Ag(NH_3_)_2_]NO_3_ followed by calcination in air was used to prepare the Ag catalysts. The citrate sol-gel method allowed preparing the ultrafine oxide materials and ensured good homogeneity through mixing of the initial components at the molecular level in solution [28,29]. This synthesis method allows obtaining the supports with the required elemental and phase composition, optimal specific surface area and pore size distribution, structural and textural characteristics [30,31]. In turn, the use of [Ag(NH_3_)_2_]NO_3_ as a silver precursor to prepare the Ag catalysts was expected to ensure a strong interaction between the Ag precursor and support resulting in stabilization of silver cations (Ag^+^) and/or oxidized silver clusters (Ag_n_^δ+^) as active species on the catalyst surface [13,32]. The obtained catalysts were characterized by such methods as X-ray fluorescence (XRF) analysis, N_2_ adsorption/desorption, Raman spectroscopy, transmission electron microscopy with analysis of the surface composition by X-ray energy dispersive spectroscopy (TEM/EDX) and X-ray photo-electron spectroscopy (XPS) and were studied in the NO SCR with C_3_H_6_ cofeeding H_2_ in the reaction mixture. For comparison, the catalytic activity in terms of NO conversion and selectivity towards N_2_ was also investigated for the supports only.

## 2. Materials and Methods

### 2.1. Preparation of the Supports

The individual CeO_2_, MnO_x_ and binary CeO_2_–MnO_x_ (with a molar ratio of Ce/Mn = 1) oxide supports were synthesized by sol–gel citrate method. Analytical grade Ce(NO_3_)_3_·6H_2_O and Mn(NO_3_)_2_·6H_2_O salts (Unihim, St. Petersburg, Russia) were used as Ce and Mn precursors, respectively, and citric acid C_6_H_8_O_7_·H_2_O (Khimprom, Kemerovo, Russia) was employed as a chelating agent. All reagents were used directly without any further purification. The colloidal solutions to synthesize the supports were prepared in a 500 mL ceramic tank using a heated magnetic stirrer. For this, the required volume of solutions of the corresponding metal precursors was rapidly added to a citric acid solution at a vigorous stirring with the molar ratio C_6_H_8_O_7_·H_2_O/(Me) = 1.2 (pH of solution was ~1–2) followed by heating up to 70 °C at a constant stirring. The above colloidal solutions were hold at 70 °C at a constant stirring for 2 h followed by the gel formation. To age the gel and additionally evaporate water, the resulting gel was placed in a drying oven overnight at 80 °C. The resulting gel was additionally dried at 120 °C (a heating rate was 10 deg/min) for 5 h and then calcined at 500 °C for 3 h with a linear heating rate up to a maximum set temperature of 5 °C/min. The synthesized samples were labelled as follows: CeO_2_, MnO_x_ and CeMnO_x_.

### 2.2. Preparation of Supported Ag Catalysts

Based on the obtained supports, a series of Ag catalysts with a fixed silver content (1 wt.%) was prepared by incipient wetness impregnation using an aqueous solution of ammonium silver complex [Ag(NH_3_)_2_]NO_3_ as an Ag precursor. The wetness of the support was determined by adding drop by drop a known volume of water solution. The volume and concentration of the impregnating solution for each support were fixed taking into account its wetness and weight to ensure 1 wt.% of Ag in the final catalyst. The samples impregnated were dried at 70 °C and then calcined at 500 °C for 2 h. The obtained catalysts were designated as follows: Ag/CeO_2_, Ag/MnO_x_ and Ag/CeMnO_x_.

### 2.3. Aging of Supported Ag Catalysts

Samples were aged thermally. Sample powders were heated in air to 650 °C at a heating rate of 10°/min, calcined at 650 °C for 12 h, and then cooled to room temperature.

### 2.4. Characterization 

The prepared samples were studied by several characterization methods, including X-ray fluorescence analysis (XRF), nitrogen adsorption/desorption at −196 °C, X-ray analysis (XRD), Raman spectroscopy, transmission electron microscopy (TEM) with analysis of the surface composition by X-ray energy dispersive spectroscopy (EDX) and X-ray photoelectron spectroscopy (XPS).

#### 2.4.1. X-ray Fluorescence Analysis

The chemical composition of the samples was analyzed using the XRF-1800 sequential X-ray fluorescence spectrometer (Shimadzu, Tokyo, Japan).

#### 2.4.2. Low-Temperature Nitrogen Adsorption/Desorption

The specific surface area, total pore volume and average pore diameter were determined from the low-temperature nitrogen adsorption/desorption (at −196 °C) using the TriStar II 3020 specific analyzer (Micromeritics, Norcross, GA, USA). Prior to experiments, all samples were degassed at 200 °C in a vacuum (10^–2^ Torr) for 2 h using the laboratory degassing station VacPrep Degasser (Micromeritics). The specific surface area was determined by the Brunauer-Emmett-Teller (BET) method; the pore volume and pore size distributions were determined by the Barrett-Joyner-Halenda (BJH) method using the desorption branch of the adsorption-desorption isotherm.

#### 2.4.3. XRD

The XRD patterns for the samples were recorded by the X-ray diffractometer XRD-7000 (Shimadzu) with monochromatic CuKα radiation (1.54 Å) in the angle range of 10–70° 2θ and a scanning rate of 0.02 °/s. The data were obtained using the Bragg-Brentano geometry. Crystalline Si (a = 5.4309 Å, λ = 1.540562 Å) was used as an external standard to calibrate the diffractometer. The phase composition was analyzed using the PDF-2 database (Release 2012 RDB). To refine the lattice parameters and determine the crystalline size, the POWDER CELL 2.4 full profile analysis program was used.

#### 2.4.4. Raman Spectroscopy

Raman spectra were obtained on the InVia spectrometer (Renishaw, UK) equipped with the DM 2500M microscope (Leica, Germany) with a 50× objective. For excitation, the lasers with wavelengths of 532 and 785 nm and a power of 100 mW were used; the spectral resolution was 2 and 1 cm^−1^, respectively. To prevent changes in the samples, only 5% of the full laser power and a 50% beam defocusing were applied.

#### 2.4.5. TEM and EDX

Transmission electron microscopy (TEM) data were obtained using the double aberration-corrected (Thermo Fisher Scientific Themis Z, Netherlands) electron microscope operated at 200 kV. Images in Scanning-TEM (STEM) mode were taken using the high-angle annular dark field (HAADF) detector. The local composition of the samples was studied using the Thermo Fisher Scientific Super-X EDX spectrometer. The samples for the TEM study were dispersed ultrasonically and deposited on copper grids covered with a holey carbon film.

#### 2.4.6. XPS

The samples were analyzed by X-ray photoelectron spectroscopy (XPS) using the photoelectron spectrometer ES 300 (Kratos Analytical, UK). Mg Kα (hν = 1256.6 eV) and Al Kα (hν = 1486.6 eV) X-ray sources were employed to acquire photoelectron spectra. To perform XPS analysis, the samples were fixed on a sample holder using a scotch-tape. The core-level spectra, namely, Ag3d, O1s, C1s, Ce3d, Mn2p, Mn3s, and Auger spectra for silver Ag MNN were acquired to estimate the quantitative composition of the samples surface as well as to analyze the oxidation state of the elements. The C1s line of the residual amorphous carbon species with a binding energy E_b_(C1s) = 285.1 eV was used as an internal standard to calibrate the spectra. Such a calibration procedure gave the E_b_ value of the U’’’ component of the Ce3d spectrum as E_b_(U’’’) = 916.7 eV being consistent with the literature data for ceria-based catalysts. The spectra were analyzed after Shirley background subtraction. The Ce3d, Mn2p, and Mn3s spectra were fitted with a combination of Gaussian and Lorentzian functions. The XPS-Calc program [33,34] was used for spectra processing. Atomic ratios were calculated using the area of the corresponding peaks with the consideration of the atomic sensitivity factor for each element [35].

### 2.5. Activity Tests in C_3_H_6_-SCR of NO

All the C_3_H_6_-SCR tests were performed in a fixed-bed continuous-flow U quartz reactor with an inner diameter of 12 mm. The feed gas consisting of 1000 ppm NO + 3600 ppm C_3_H_6_ + 2.9 vol.% H_2_ + 10 vol.% O_2_ in He was flowed over the catalyst (120 mg) at a rate of 50 mL·min^−1^ equivalent to a weight hourly space velocity (WHSV) of 25,000 mL g^−1^ h^−1^. To study the effect of hydrogen presence, catalytic tests without it, where also carried out. To this purpose the feed gas consisting of 1000 ppm NO + 3600 ppm C_3_H_6_ + 10 vol.% O_2_ in He was used. The conversion values were measured as a function of temperature from 100 °C to 500 °C with a heating rate of 5 °C/min, holding 40 min at each temperature that was increased by steps of 50 °C. The inlet and outlet gas compositions were analyzed by mass quadrupole spectrometer (ThermostarTM, Balzers, Liechstenstein) and by ABB detectors, infrared (Limas 11) for NO, N_2_O, NO_2_, paramagnetic (Magnos 206) for O_2_, and UV (Uras 14) for CO and CO_2_ detection. 

The NO conversion, selectivity to N_2_ and N_2_ yield were calculated according to procedures described in Refs. [9,12]:$$NO\;\mathrm{conversion}\;\left(\%\right):\frac{{\left[NO\right]}_{in}-{\left[NO\right]}_{out}}{{\left[NO\right]}_{in}}\times 100;$$
$$\mathrm{Selectivity}\;\mathrm{to}\;N_{2}\;\left(\%\right):1-\frac{{\left[NO_{2}\right]}_{out}+2{\left[N_{2}O\right]}_{out}}{{\left[NO\right]}_{in}-{\left[NO\right]}_{out}}\times 100;$$
$$N_{2}\;\mathrm{yield}\;\left(\%\right):\;\frac{{\left[NO\right]}_{in}-{\left[NO\right]}_{out}-2{\left[N_{2}O\right]}_{out}-{\left[NO_{2}\right]}_{out}}{{\left[NO\right]}_{in}}\times 100.$$

## 3. Results and Discussion

### 3.1. Chemical Composition and Textural Characteristics of the Samples

Table 1 shows the XFR data for the prepared Ag catalysts. The composition of the supports is consistent with those of the catalysts and is not presented in the table. According to the data obtained, there is a good consistency between the real and nominal chemical compositions.

Figure 1 shows nitrogen adsorption-desorption isotherms and pore size distributions for CeO_2_, MnO_x_ and CeMnO_x_ supports and the corresponding Ag catalysts. Table 2 lists the values of specific surface area and total pore volume. For all samples, the adsorption–desorption isotherms belong to type IV according to the IUPAC classification [36]. The observed hysteresis loops of type H_2_ indicate the presence of mesopores with a complex structure in the samples. For the CeO_2_ and CeMnO_x_ samples, wide hysteresis loops are observed in the relative pressure range of 0.45–1.0 corresponding to a rather narrow pore size distribution in the range of 2–10 nm. For the MnO_x_ sample, a narrow hysteresis loop is observed in the relative pressure range of 0.82–1.0, which corresponds to a wide pore size distribution in the range of 4–200 nm with a maximum at ~30 nm. The CeO_2_ and CeMnO_x_ supports are characterized by the relatively high specific surface area (40 and 51 m^2^/g, respectively) and total pore volume (0.140 and 0.222 cm^3^/g, respectively), while the MnO_x_ sample shows relatively low specific surface area (14 m^2^/g) and total pore volume (0.115 cm^3^/g).

The Ag introduction does not significantly affect the isotherms and pore size distributions in the corresponding samples (Figure 1). The observed decrease in the specific surface area with a slight change in the total pore volume is apparently due to the support sintering according to the mechanism of surface diffusion during the calcination step. In general, the changes observed in the textural characteristics of the Ag/oxide catalysts indicate a uniform distribution of Ag introduced into the porous space of the supports.

### 3.2. Phase Composition and Structural Characteristics of Samples

The phase composition and structural features of the CeO_2_, MnO_x_ and CeMnO_x_ supports, and the corresponding Ag catalysts were studied by XRD, Raman spectroscopy with laser excitation wavelengths of 532 and 785 nm, and TEM. Figure 2 shows the XRD patterns for the samples studied. Table 2 presents the phase composition of the samples and the characteristics of the crystalline phases revealed during the XRD data analysis. Figure 3 shows the Raman spectra for the samples at two different laser excitation wavelengths; the use of various laser excitations provides wide information due to the resonance effect of Raman scattering [37,38]. Figure 4, Figure 5 and Figure 6 display the high-resolution TEM (HRTEM) images and high resolution EDX mapping for Ag supported catalysts.

#### 3.2.1. CeO_2_ and Ag/CeO_2_ Samples

According to the XRD data, the CeO_2_ sample has a cubic fluorite-type phase with the lattice parameter *a* of 5.405 Å and the average XRD crystallite size of 13 nm. The formation of fluorite-type ceria is confirmed by Raman spectroscopy, with the structural defects being additionally indicated. Specifically, Raman spectrum for the CeO_2_ support obtained under 532 nm laser contains the bands at 260, 404, 464, 560, 590 and 828 cm^−1^ (Figure 3a), while the one obtained under 785 nm laser is characterized by the bands at 276, 314, 410, 464, 556, 595, and 724 cm^−1^ (Figure 3b). The intense band at 464 cm^–1^ is attributed to the F2g band associated with the Ce–O stretching vibration in the [CeO_8_] cubic subcell of ceria, and the weak bands located at 256–260, 404–410, and 590–595 cm^–1^ are referred to overtones [39]. The band at ~560 cm^–1^ is associated with the Ce^3+^ located in the immediate vicinity of the oxygen defect [39], the band at ~314 cm^–1^ is attributed to the displacement of oxygen atoms from the ideal positions of the fluorite lattice [38], and the band at 828 cm^–1^ is attributed to the surface peroxide O_2_^2−^ species [39,40]. The presence of these bands indicates a defective structure of cerium oxide obtained by the citrate method.

For Ag/CeO_2_ sample, no additional Ag-based phases were revealed by XRD, which is caused by the low Ag content as well as high dispersion of such phases. The lattice parameter *a* of the fluorite-type phase remains unchanged (5.405 Å), and the average crystallite size is 15 nm. These results are additionally confirmed by the HRTEM data (Figure 4) indicating the ceria crystallites of ~5 to 20 nm in size for the Ag/CeO_2_ catalyst. According to the high resolution EDX analysis, silver is evenly distributed over the sample, however, in some places silver is concentrated as oxide species, which is reduced under the electron beam. The Raman spectroscopy does not reveal a significant effect of the Ag deposition on the ceria structure in the Ag/CeO_2_ sample. Thus, a specific absorption below 200 cm^−1^ due to the Ag–O–Ce bond vibrations additionally appears in the spectra for the sample (Figure 3), while other bands remain intact.

#### 3.2.2. MnO_x_ and Ag/MnO_x_ Samples

According to the XRD results, the MnO_x_ sample is characterized by the presence of two crystalline phases, namely, the tetragonal Mn_3_O_4_ and the cubic Mn_2_O_3_ in amounts of ~75 wt.% and ~25 wt.%, respectively (Figure 2, Table 2). The oxide phase crystallites formed in the MnO_x_ sample are noticeably larger than in the case of the CeO_2_ sample. The Mn_2_O_3_ phase is rather well crystallized and characterized by the average XRD crystallite size of 61 nm. The Mn_3_O_4_ phase is less ordered and more dispersed, with the average crystallite size being 33 nm. The Raman spectroscopy data confirm the formation of Mn_3_O_4_ and Mn_2_O_3_ oxides and additionally reveal the presence of some MnO oxide in the MnO_x_ sample. Thus, the Raman spectrum for the MnO_x_ support obtained under 532 nm laser (Figure 3a) is characterized by an intense band at 647 cm^−1^ and weak bands at 263, 308, 360 and ~480 cm^−1^ characteristic of Mn_3_O_4_ with the spinel structure [41,42,43,44]. The intense band at 647 cm^–1^ is attributed to the A1g mode of the Mn–O stretching vibration for Mn^2+^ ions in tetrahedral coordination, and the weak bands at 263, 308, and 360 cm^–1^ are attributed to the Eg, A1g, B2g, and Eg modes, respectively [42]. The spectrum also contains week bands in the ranges of 150–240 cm^–1^ and 500–600 cm^–1^, which are caused by other manganese oxide phases, i.e., Mn_2_O_3_ and Mn_5_O_8_. These phases are more reliably distinguished in the spectrum obtained under 785 nm laser (Figure 3b) due to the resonance effect of Raman scattering resulting in the presence of well-defined bands at 170, 262, 393, 477, 535 and 580 cm^–1^ attributed to Mn_5_O_8_ modes [45], and the bands at 191, 311 and 621sh cm^–1^ attributed to Mn_2_O_3_ [43,46,47], with those at 286, 367, and 648 cm^−1^ being assigned to Mn_3_O_4_ with a spinel structure [42,44]. The amount of the Mn_5_O_8_ phase seems to be negligible and is determined by Raman spectroscopy due to the high extinction coefficient.

Signs n.a. means not available due to the correct determination of a full width at half maximum is impossible.

The Ag introduction is accompanied by the increase in the relative Mn_2_O_3_ content up to 76 wt.% and the decrease in that of Mn_3_O_4_ up to 24 wt.% in the Ag/MnO_x_ sample according to the XRD results. Besides, the Ag introduction results in the appearance of well-defined bands at 196, 308 and 625sh cm^−1^ attributed to Mn_2_O_3_ and at 170, 263, 533 and 576 cm^–1^ attributed to MnO in the spectrum for the Ag/MnO_x_ sample obtained under 532 nm laser (Figure 3a). In the spectrum obtained under 785 nm laser (Figure 3b), the relative intensities of the bands of Mn_2_O_3_ and MnO increase, with the additional bands assigned to Mn_2_O_3_ being distinguished at 191, 393, and 697 cm^−1^, which is consistent with the increase in the Mn_2_O_3_ content in the sample according to the XRD data. The average XRD crystallite size for the main Mn_2_O_3_ phase is 52 nm, which is consistent with the presence of Mn_2_O_3_ particles revealed by TEM in the Ag/MnO_x_ sample (Figure 5), with their size varying from 20 to 100 nm. Besides, according to high resolution EDX mapping, the silver distribution in the sample is less uniform than for the CeO_2_ sample. The mapping shows areas depleted (0.15–0.59 wt.%) and enriched (3–36 wt.%) with silver.

Some areas enriched with silver correspond to Ag particles (Figure 5, region II), however, most other areas enriched with silver correspond to oxidized silver species (Appendix A) in accordance with the XPS data (see Section 3.3 for details).

#### 3.2.3. CeMnO_x_ and Ag/CeMnO_x_ Samples

The XRD data for the CeMnO_x_ sample indicate the presence of only cubic phase with the fluorite structure, while individual phases of manganese oxides (MnO_2_, Mn_2_O_3_, Mn_3_O_4_, or MnO) are not found. According to the XRD data, the average crystallite size of the fluorite-type phase is 7 nm in the CeMnO_x_ sample, which is almost two times smaller than in the CeO_2_ sample. At the same time, the parameter *a* of the fluorite phase of 5.406 Å indicates the presence of CeO_2_ phase rather than the Ce_1-x_Mn_x_O_2_ solid solution with the fluorite structure, which should be characterized by a noticeable compression of the crystal lattice due to the substitution of Ce^4+^/Ce^3+^ ions by smaller Mn^4+^/Mn^3+^ ions [48,49]. This finding was confirmed by the Raman spectroscopy and TEM data. The Raman spectra for CeMnO_x_ sample contain two intense broad bands with maxima at 454 and 644 cm^−1^ in the case of 532 nm laser and at 459 and 647 cm^−1^ in the case of 785 nm laser. In both cases, absorption below 400 cm^−1^ is additionally observed as well as the additional shoulder peaks at 495, 410, 537, 553, 590, and 620–625 cm^−1^ can be distinguished. The indicated bands are caused by the individual CeO_2_ and Mn_3_O_4_/Mn_2_O_3_ oxides. Such finding is consistent with the formation of undoped CeO_2_ according to the XRD data analysis. A strong broadening of the bands suggests a high dispersion and/or distortion of the oxide phases presented in the CeMnO_x_ sample in consistency with the HRTEM results indicating the formation a “patchwork” domain microstructure with rather small crystallite with sizes from 1.5 to 3 nm enriched by either Mn or Ce (Figure 6). The Mn/Ce atomic ratio is ~3/1 in some domains and ~1/2 in other domains. The interplanar spaces of ~0.33 and ~0.28 nm are primarily observed for both domain types that are typical for CeO_2_ fluorite-type (JCPDS 34–0394) and cubic α-Mn_2_O_3_ bixbyite structures (JCPDS 41–1442), with the latter being the oxygen-deficient fluorite-related structure.

The Ag introduction does not affect the crystallite size of fluorite phase and does not lead to the appearance of additional crystalline phases in the Ag/CeMnO_x_ sample. At the same time, the Ag introduction leads to a slight increase in the parameter *a* of the fluorite-type structure up to 5.413 Å in the Ag/CeMn sample (Table 2). This finding can be attributed to the formation of Ce^3+^ ions characterized by larger ionic radius (i.r.) (i.r. = 1.28 Å for CN = 8) than for Ce^4+^ ions (i.r. = 0.97 Å for CN = 8 [32]). The latter can result from the “bulk oxygen pump out” effect caused by the reverse spillover of oxygen from CeO_2_ to Ag [50,51]. The Raman spectroscopy data reveal that the Ag introduction results in some changes in the range of 500–700 cm^−1^, which confirms the Ce^3+^ formation in ceria (appearance of the band at ~553 cm^−1^ assigned to the defect-induced D1 mode) as well as indicates the change in the relative content of Mn_3_O_4_/Mn_2_O_3_ manganese oxides in the Ag/CeMnO_x_ sample. According to the high resolution EDX mapping, silver is rather evenly distributed throughout the sample, with the XPS data indicating the primarily formation of Ag^+^ ions dispersed on the surface or subsurface of the oxide support matrix (see Section 3.3 for details). In some parts, isolated silver nanoparticles are observed (Appendix A).

Therefore, the use of the citrate method to prepare the CeMnO_x_ support provides the formation of fluorite-type oxide nanocomposite with the “patchwork” nanodomain microstructure. This is caused by a good homogeneity achieved through mixing of the initial components at the molecular level in the gel formed, with the limited solubility of cerium and manganese oxides resulting in the system disintegration under thermal treatment in air to form nanodomains enriched with either Mn or Ce.

### 3.3. Surface Composition of the SUPPORTEd Ag Catalysts

The surface composition of the Ag catalysts was additionally studied by XPS. Table 3 shows atomic ratios of the elements on the sample surfaces. According to the XPS data, the Ce/Mn as well as Ag/Ce, Ag/Mn, and Ag/(Ce+Mn) atomic ratios on the surface of the Ag/CeMnO_x_ sample correspond to the nominal ones according to XRF data. This indicates the uniform distribution of Ce, Mn and Ag in the sample, which is consistent with the oxide support microstructure formed by the 1.5–3 nm nanodomains enriched with either Mn or Ce and even Ag distribution over the sample revealed by HRTEM and EDX. For the Ag/CeO_2_ sample, the Ag/Ce surface atomic ratio is also rather close to the nominal one, but the Ag/Mn surface atomic ratio for the Ag/MnO_x_ sample is significantly lower than the nominal value. The observed deviation of the surface Ag content is associated with the formation of rather large silver oxide particles (10–50 nm) in the samples (Appendix A).

#### 3.3.1. Oxidation States of Cerium and Manganese

Figure 7 shows the Ce3d spectra for Ag/CeO_2_ and Ag/CeMnO_x_ samples. Based on the literature data [52], the Ce 3d spectra were fitted with several peaks corresponding to Ce^4+^ (V, V″,V‴ and U, U″, U‴ peaks) and Ce^3+^ (V_0_,V′ and U_0_,U′ peaks) species. The relative fraction of Ce^3+^ species calculated as a ratio of the V_0_,V′ and U_0_,U′ peak areas to the one of the overall Ce3d peak was ~10% for both samples.

To correctly interpret the manganese charging state, the Mn2p and Mn3s spectra were analyzed (Figure 8). The Mn2p_3/2_ peak maximum is characterized by E_b_(Mn2p_3/2_) = 641.4 eV. Such E_b_ is often observed for Mn_2_O_3_ and Mn_3_O_4_ oxides [53,54]. Analysis of the splitting between the Mn2p_1/2_ peak maximum and shake-up satellite gives ΔE_sat_ = 10.3 eV. Such ΔE_sat_ value also indicates the formation of Mn_2_O_3_ and Mn_3_O_4_ oxides [55]. Analysis of the multiplet splitting of the Mn3s spectra was also used to identify the Mn oxidation state. For Ag/Mn and Ag/CeMnO_x_ samples, the ΔE is ~5.4–5.6 eV (Figure 8b). According to the literature data, such Mn3s multiplet splitting is typical for Mn_2_O_3_ and Mn_3_O_4_ oxides [53,54,55]. Thus, analysis of the Mn2p and Mn3s core-level spectra indicates the preferential formation of Mn^3+^ species in the composition of Mn_2_O_3_ and/or Mn_3_O_4_ oxides.

#### 3.3.2. Oxidation State of Silver 

To analyze the Ag oxidation state in the samples, the core-level Ag3d spectra and Auger spectra AgMNN were collected (Figure 9). The Ag3d spectra for all samples can be fitted with one Ag3d spin-orbit doublet peak with a binding energy of the Ag3d_5/2_ peak E_b_(Ag3d_5/2_) being in the range of 367.6–368.1 eV. The E_b_(Ag3d_5/2_) = 367.6 eV is usually considered characteristic of the oxidized silver species, while the E_b_(Ag3d_5/2_) values of ~368.0 and 368.1 eV are related to the metallic silver species [56,57]. However, it is known that the exact E_b_(Ag3d_5/2_) value is rather sensitive to the size of the silver particles, their interaction with the support and possible charging effects. To get reliable data on the oxidation state of silver in the samples, the modified Auger parameter (α’) calculated as a sum of the binding energy of Ag3d_5/2_ peak and the kinetic energy of M_4_N_4,5_N_4,5_ Auger peak were considered. Figure 7b shows the corresponding Auger spectra. Analysis of the α’ values (Table 3) indicates that in the Ag/MnO_x_ sample, silver exists in the oxidized state similar to the one in Ag_2_O species [56,57]. Ag/CeO_2_ and Ag/CeMnO_x_ samples are characterized by the lower α’ value. Such α’ value was detected for the Ag^+^ ions in the composition of inorganic salts [33,57]. Thus, the formation of Ag^+^ ions dispersed on the surface or in the subsurface region of the oxide support matrix can be proposed. The ratio of the AgMNN and Ag3d peak areas is rather similar for all samples. Slightly higher AgMNN/Ag3d value for the Ag/CeO_2_ sample might indicate higher degree of surface localization of the silver species.

### 3.4. Catalytic Performance

The obtained catalysts and the related support oxides were investigated in the NO SCR with C_3_H_6_ in the temperature range from 100 to 500 °C using H_2_ 2.9 vol.% as a reductant in the reaction mixture. All the figures herein reported are related to catalytic tests carried out in presence of hydrogen in the reaction mixture, unless differently specified. 

Figure 10 shows the results of HC-SCR with propene over the Ag supported catalysts. The NO conversion over these catalysts declined in as follows: Ag/CeMnO_x_ (44%) > Ag/MnO_x_ (37%) > Ag/CeO_2_ (28%).

In the C_3_H_6_-SCR process, the Ag/CeMnO_x_ catalyst shows a sharp increase in NO*_x_* conversion from 250 °C, reaching maximum NO*_x_* conversion of 44% at 300 °C, and it is much active as compared to the Ag/MnO_x_ and Ag/CeO_2_ catalysts (Figure 10a). Subsequently, the NO*_x_* conversion gradually decreases at 350 °C. This indicates that the combination of Ce and Mn oxides plays an important role in improving the HC-SCR activity at low temperatures. Figure 10b,c show the selectivity to N_2_ and N_2_ yield in C_3_H_6_-SCR: the Ag/MnO_x_ catalyst exhibits the highest selectivity to N_2_ in the whole temperature range (250–500 °C) reaching values around 97–98% between 300–500 °C, while the N_2_ yield that was close to 35%, at 300–350 °C, decreases with increasing temperature (>350 °C) indicating that undesired products such as N_2_O are formed faster at higher temperatures. Ag/CeMnO_x_ shows similar trend to Ag/MnO_x_ as for the N_2_ selectivity at 250–300 °C, achieving values close to 90%. However, it suffered a fast decline of N_2_ yield at T ≥ 350 °C. Ag/CeO_2_ is the worst catalyst in terms of NO conversion and selectivity to N_2_.

In the temperature range of 175–500 °C, a high contribution due to the C_3_H_6_ oxidation is observed for all the catalysts, with more than 80% of conversion being achieved at 175 °C for the Ag/CeMnO_x_. By comparing the NO and C_3_H_6_ conversion curves, it emerged that the propene oxidation by the oxygen present in the reaction mixture occurs alongside the C_3_H_6_-SCR of NO and above 350 °C (when the NO conversion declines) becomes the main reaction.

For comparison reasons, the C_3_H_6_-SCR of NO was also investigated for the CeO_2_, MnO_x_ and CeMnO_x_ supports (Figure 11).

According to the results, CeMnO_x_ was the most active among the investigated oxides, although the NO conversion is relatively low in the absence of Ag particles, as can be seen in Figure 11a. The values registered for CeO_2_ and MnO_x_ are even lower than those for CeMnO_x._ A similar trend was observed as far as N_2_ selectivity and yield. Therefore, the results imply that Ag loading significantly improves the NO SCR. 

Figure 10d and Figure 11d show the C_3_H_6_ conversion curves in C_3_H_6_-SCR at different temperatures for the samples studied. For the CeMnO_x_ and MnO_x_ samples, the C_3_H_6_ conversion steadily increases between 150 and 200 °C reaching 100% at 250–300 °C (Figure 11d). For CeO_2_ a sharp increase occurs at 300–400 °C and a total C_3_H_6_ conversion is achieved only at 500 °C. Conversely, as above mentioned, the silver catalysts exhibit higher activity than the corresponding supports reaching ~100% conversion at 250–350 °C (Figure 10d). It is worth noting that in all cases, for Ag catalysts and the supports, CO_2_ was the main product detected by the C_3_H_6_ oxidation with negligible amounts of CO (less than 1–2%), according to the carbon mass balance. However, we cannot exclude that secondary products, such as acetic acid (see below Figure 1), can be formed on the catalyst surface and they are fast oxidized to CO_2_. 

An important aspect to consider is the presence of H_2_ in the NO-C_3_H_6_ reaction mixture [27]. In fact, it is well known that the addition of H_2_ determines a promoting effect on the C_3_H_6_-SCR (“hydrogen effect”) [58]. Such effect results in an increased percentages of strongly adsorbed and decomposed nitrates on the catalyst surface and in the conversion of these adsorbed species into –NCO and –CN, which are supported to be the key surface intermediates for the HC-SCR reaction [59,60]. 

In this respect, the reaction mixture containing 2.9% of hydrogen was used as a standard. To study the effect of hydrogen addition, the catalytic properties of Ag/CeMnO_x_ and Ag/MnO_x_ samples, as the most perspective, were studied using H_2_-free reaction mixture. Figure 12 shows the results obtained. The activity and selectivity of the Ag/MnO_x_ catalyst were notably lower without hydrogen in all temperatures studied (see Figure 10 for comparison). Whereas, the Ag/CeMnO_x_ catalyst shows similar catalytic performances in the absence of hydrogen as compared with those in the presence of hydrogen.

To study the catalyst stability, the samples were thermally aged. Figure 13 shows the results on SCR study using standard reaction mixture obtained for aged samples. According to the data obtained, the thermal aging results in deactivation of the Ag/CeO_2_ sample and notable decrease in catalytic efficiency of the Ag/MnO_x_ sample, while the performances of the Ag/CeMnO_x_ catalyst was practically unchanged, which was assigned to a rather high stability of its phase composition and textural characteristics as compared with those of Ag/CeO_2_ and Ag/MnO_x_ samples (for details see Appendix A).

Thereby, the results obtained indicate that the 1%Ag/CeMnO_x_ showed the highest catalytic efficiency in both catalytic properties and stability among studied catalyst, with its activity and selectivity being comparable or superior as compared with those previously reported for 1%Ag/CeZrO_x_ catalyst (Table 4) [27]. Thus, the 1%Ag/CeMnO_x_ showed comparable NO conversion of 46% and significantly higher N_2_ selectivity of 86% as compared with 50% NO conversion and 30% N_2_ selectivity 1%Ag/CeZrO_x_ under H_2_-free conditions those resulting in superior overall efficiency (40% N_2_ yield vs 15% N_2_ yield). The hydrogen addition in the reaction mixture results in notable improvement of both activity and selectivity of the 1%Ag/CeZrO_x_ catalyst, with the NO conversion increasing up to 80%, N_2_ selectivity increasing up to 60%, and N_2_ yield increasing up to 48%. The effect of hydrogen addition on the 1%Ag/CeMnO_x_ catalyst performance was less noticeable, with the NO conversion decreasing up to 44%, N_2_ selectivity increasing up to 91%, and N_2_ yield remaining 40%. Despite the slightly inferior overall efficiency of the 1%Ag/CeMnO_x_ catalyst in these conditions, it showed high stability after aging, which makes it promising for further CH-SCR catalyst development.

As previously discussed and reported in literature [58], different Ag species such as isolated silver cations (Ag^+^), oxidized silver clusters (Ag_n_*^δ^*^+^), and metallic silver clusters (Ag_n_^0^) can been observed in the Ag catalysts and HC-SCR catalysts; the oxidized silver species (Ag^+^ and/or Ag_n_*^δ^*^+^) play an important role, in fact they are proposed to be the active species in the NO-SCR reaction with propene, whereas the Ag_n_^0^ metallic clusters are responsible for the nonselective oxidation of hydrocarbons. Thus, NO adsorption with dimer formation was shown to be favourable on the supported silver Ag_2_^+^ clusters followed by its reduction with HC or alcohol to form N_2_ and N_2_O [61]. The NO reduction activity was clarified to be controlled by partial oxidation of C_3_H_8_ mainly to surface acetates [59]. Figure 1 presents the proposed scheme of NO reduction on Ag_n_*^δ^*^+^, specifically Ag_2_^+^, clusters based on the literature data.

Based on the so far reported literature, the catalytic performance of supported Ag catalysts is controlled by many factors, including morphological, structural and electronic ones. In the present work, the citrate sol-gel method has produced a mixed oxide CeMnO_x_ (with molar ratio Ce/Mn = 1) with “patchwork” nanodomain microstructure, where silver is uniformly distributed as Ag^+^ and/or Ag_n_*^δ^*^+^ species that are supposed provide NO adsorption to form N_2_O_2_ dimers that are subsequently reduced with propylene to form N_2_ at low temperatures. At high temperatures (>300 °C), the competitive total propylene oxidation seems to lead to the decrease in the catalyst activity in the NO reduction.

## 4. Conclusions

The Ag/CeO_2_, Ag/MnO_x_, and Ag/CeMnO_x_ catalyst with 1 wt.% Ag were successfully prepared using a combination of citrate sol-gel method for support synthesis and incipient wetness impregnation with [Ag(NH_3_)_2_]NO_3_ aqueous solution to deposit the active component. The used approaches provided the formation in the CeMnO_x_ of a characteristic “patchwork” domain microstructure that along with the presence of well dispersed Ag^+^/Ag_n_*^δ^*^+^ species strongly interacting with the support, produced a catalyst with higher NO-SCR performance and perfect stability compared to Ag/CeO_2_ and Ag/MnO_x_ systems, achieving 44% NO conversion at 300 °C under a WSHV of 25,000 mL g^−1^ h^−1^ and selectivity to N_2_ close to 90%. At temperatures above 300 °C, a high contribution from the C_3_H_6_ oxidation was observed for all catalysts. Since it is well agreed that for an efficient NO SCR catalyst it is required to increase the NO conversion values and to expand the temperature range of operation, it can be concluded that this achievement may be associated with the investigation of different Ag loadings and as well with the decrease of CeO_2_ molar fraction in a CeMnO_x_ mixed oxide, while maintaining its microstructure.

## Data Availability

No applicable.

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
