# Peer review of "Ag Catalysts Supported on CeO2, MnO2 and CeMnOx Mixed Oxides for Selective Catalytic Reduction of NO by C3H6"

_nanomaterials, 2023, doi:10.3390/nano13050873_

Round 1

Reviewer 1 Report

1.        The SCR should be provided in full form in the title.

2.        Reduce the use of abbreviations to bare minimum in the abstract. Provide the full form of most of the abbreviations used in the abstract.

3.        Impregnation of silver by what method – please provide in the abstract.

4.        “Sol-gel method” repeated twice in the second line of the abstract.

5.        The quantified data on catalytic efficiency should be included in the abstract.

6.        When there are several characterization tools used, why only HRTEM is mentioned in the keywords?

7.        All the abbreviations used in the manuscript should be defined in full form at the first instance and abbreviated thereafter. Please double-check throughout the manuscript.

8.        All the purchase details of chemicals/reagents and instruments/equipment/software/kits should be provided as state, city and country in the case of USA as well as city and country in the case of other countries. Also, for the second instance of same vendor/company’s mention, the authors can simply mention the company name, for instance as Sigma-Aldrich and not very time Sigma-Aldrich (USA).

9.        Provide a reference citation section 2.2 procedure.

10.     The equations in page 4 doesn’t need the % sign on the right-hand side.

11.     All the abbreviations used in figures (including variables used in x-axis and y-axis) and tables should be described in full form under the respective figure caption and table footnote.

12.     The catalytic efficiency reported in this study should be compared with those reported for similar metal oxides. A comparative table should be prepared and data discussed to claim the superiority of the catalytic efficiency shown in this article.

13.     Conclusion should of one paragraph.

Author Response

We are very grateful to the reviewer for comments and suggestions. Please, attached you can find the detailed responses. 

Reviewer 2 Report

The manuscript presents a detailed study on the catalytic activity of Ag/CeO2, MnO2, and mixed CeMnOx with Ce/Mn=1 in NOx SCR with C3H6. The authors have performed an extensive characterization of the supports and catalysts using advanced characterization techniques. Considering the modest results in C3H6-SCR of NOx (maximum conversion level of 44%), the paper is interesting to read mostly for the characterization results. 

However, the presentation of the characterization results in section 3.2 is somehow difficult to follow since the discussions are made in some cases (Ag/MnOx) rather far from the figures and tables they are refering to. An easier to follow approach would be to comment comparatively the results obtained by each characterization technique.

Also, it would be interesting if the authors could attempt to explain why there is the change in the crystallographic phases of MnOx support (which was already calcined) after it was impregnated with [Ag(NH3)2]NO3 solution. This fact would suggest a possible dissolution-recrystallization phenomenon during the impregnation.

Other aspects that should be improved are the following:

1) in the introduction section the authors should emphasize the original aspects related to their research considering the fact that they wrote at lines 76-79 : "According to the available literature data, the Ag catalysts supported on Ce, Mn and Ce-Mn mixed oxides are appealing as catalytic materials for NOx SCR in exhausts emitted by diesel engines of vehicles and ships in compliance with the EURO VI and IMO 2020 regulations [25]"

2) the value of WHSV in the abstract line19 is written 25.000 mLxg-1xh-1 while at line 183 it is 25,000mLxg-1xh-1 - which value is correct?

3) In section 2.2. line 119 - the authors state that the supported silver catalysts were obtained by "incipient wetness impregnation using the minimum volume of aqueous solution of ammonium silver complex" - the minimum volume should be defined in order to give the possibility to other researchers to reproduce the experiment at least a range of L/S ratios shold be given and the concentration of Ag in the aqueous solution. 

Author Response

(The authors gave the same response as above.)

Reviewer 3 Report

This paper describes the synthesis of Ag catalysts using CeO2, MnO2 and CeMnOx support. The prepared catalysts were applied to NOx conversion reaction.

1.        This study does not have scientific soundness.

2.        The results do not clearly support the author’s claims. The study is baseless.

3.        The authors claimed that Ag/CeMnOx had higher efficiency than Ag/CeO2 and Ag/MnOx. Whereas, according to Figure 10, the NOx conversion has a very slight difference which can be attributed to the error. There is not any distinct difference in the efficiencies of the three catalysts. The claim of higher efficiency of Ag/CeMnOx catalyst is baseless.

4.        The characterization results do not show any strong evidence for the higher efficiency of Ag/CeMnOx catalyst. Raman results do not show any evidence for higher efficiency of Ag/CeMnOx catalyst. The XPS results are also insufficient to support the claim.

5.        Why Ce/Mn molar ratio 1 was selected. The authors should prepare catalysts in various Ce/Mn molar ratios to find out the effect of mixing.

6.        What is the role of MnOx in Ag/CeMnOx catalyst? How addition of MnOx affects the catalytic efficiency? What is the optimized rato of Ce/Mn to enhance the catalytic efficiency?

7.        What is the mechanism of reaction?

Author Response

We are grateful to the reviewer for comments and suggestions. Please, attached you can find the detailed responses. 

Round 2

Reviewer 2 Report

At line 100 of the manuscript add the word "fluorescence" after X-ray

Line 128 the authors state : The wetness of the support was determined by the drop method. (This sentence should be rephrased since until now I have never heard of the "drop method"...even if I can imagine what the authors want to say it should be explained in a satisfactory English)

Author Response

We are very grateful to the reviewer for further comments and suggestions.

Comments and Suggestions for Authors:

At line 100 of the manuscript add the word "fluorescence" after X-ray.

The word "fluorescence" has been added

Line 128 the authors state : The wetness of the support was determined by the drop method. (This sentence should be rephrased since until now I have never heard of the "drop method"...even if I can imagine what the authors want to say it should be explained in a satisfactory English).

The sentence has been rephrased as follows: “The wetness of the support was determined by adding drop by drop a known volume of water solution”.

Reviewer 3 Report

The answers to the comments are unsatisfactory. The authors keep saying that they will provide evidence in other studies or papers, and no study on the role of MnOx in the support, optimized value for Ce/Mn molar ratio and mechanism is provided in this paper. Publishing a paper without evidence and proper study on the catalyst’s characteristics is not ethical and not acceptable scientifically. Incomplete studies can provide wrong information to the scientific community.

Comment 1. This study does not have scientific soundness.

Author’s Response: We do not agree with this comment. We try to emphasize the original aspects related to the research in the introduction.

Second review report: A study without any proper evidence of claims, without providing the mechanism, role of each component of the catalyst in the reaction, and optimized Ce/Mn molar ratio in the catalyst does not have scientific soundness. The authors do not have any proper reason to describe why Ag/CeMnOx has higher efficiency than other catalysts in terms of its component. Why mixing of CeOx and MnOx gives higher efficiency. Moreover, the efficiency of Ag/CeMnOx is not so remarkably higher than other catalysts.

Comment 2. The results do not clearly support the author’s claims. The study is baseless.

Author’s Response: We do not agree with this comment. We try to emphasize this adding new results (see response on comment 3).

Second review report: A study without any proper evidence of claims, without providing the mechanism, role of each component of the catalyst in the reaction, and optimized Ce/Mn molar ratio in the catalyst does not have scientific soundness. The authors do not have any proper reason to describe why Ag/CeMnOx has higher efficiency than other catalysts in terms of its component. Why mixing of CeOx and MnOx gives higher efficiency. Moreover, the efficiency of Ag/CeMnOx is not so remarkably higher than other catalysts.

Comment 4. The characterization results do not show any strong evidence for the higher efficiency of Ag/CeMnOx catalyst. Raman results do not show any evidence for higher efficiency of Ag/CeMnOx catalyst. The XPS results are also insufficient to support the claim.

Author’s response: The methods used for sample characterization allow determining their phase composition, features of structure and morphology, a state active component. In case of the CeMnOx and Ag/CeMnOx samples, using only XRD did not allow identifying the presence of MnOx phase due to strong broadening and overlapping of the fluorite and manganese oxide lines the XRD parents. The Raman spectroscopy made it possible to reveal the formation of separate phases, rather than a solid solution based on fluorite, with TEM data helping to understand the distribution of MnOx and CeO2-based phases in the sample. The XPS results clearly show the state of silver in the sample.

Second review report: All the characterization techniques do not provide the reason for how the addition of MnOx to CeOx increased the efficiency. The reason cannot be obtained unless the authors do complete a study by preparing catalysts using various Ce/Mn molar ratios.

Comment 5. Why Ce/Mn molar ratio 1 was selected. The authors should prepare catalysts in various Ce/Mn molar ratios to find out the effect of mixing.

Author’s Response: Thank you for the comment and recommendation. The present study was focused on investigation of individual CeO2 and MnOx oxide and their equimolar mixture. The effect of Ce/Mn molar ratio as well as Ag loading on catalytic properties deserves separate consideration as it was indicated in conclusions. These issues will be considered in detail and presented elsewhere.

Second review report: It is wrong to provide incomplete information to the scientific community. The authors should check all the possible Ce/Mn molar ratios for catalyst synthesis and compare their efficiency for NOx conversion.

Comment 6. What is the role of MnOx in Ag/CeMnOx catalyst? How addition of MnOx affects the catalytic efficiency?

Author’s Response: The specific roles of the CeO2 and MnOx in activation of NO or C3H6 were not studied in this work. But both oxides are catalytically active in NOx HC-SCR, with the Ag improving the catalytic efficiency. This will be considered in detail in future studies.

Second review report: How can a paper be published without getting information about the role of each component of the catalyst? The authors should study the role of MnOx in the catalyst. How presence of MnOx in the catalyst affects the efficiency?

Comment: What is the optimized ratio of Ce/Mn to enhance the catalytic efficiency?

The present study was focused on investigation of individual CeO2 and MnOx oxide and their equimolar mixture. The effect of Ce/Mn molar ratio will be considered in detail and presented elsewhere.

Second review report: The authors must study and evaluate the optimized ratio of Ce/Mn suitable for the current study.

Comment 7. What is the mechanism of reaction?

Author’s Response: The present work was not devoted to revealing the complete reaction mechanism. Such important aspects as reaction mechanism can be a subject of future work. In this paper, we discuss possible pathways for NO conversion at silver centers (isolated silver cations (Ag+), oxidized silver clusters (Agnδ+), and metallic silver clusters (Agn0)) based on available literature data.

Second review report: The paper cannot be accepted without the necessary information. The authors should study the mechanism of the reaction using Ag/CeMnOx catalyst.

Author Response

We totally disagree with the referee's allegations and inferences and feel deeply offended. As far as we are concerned, our results are scientifically correct and with the utmost respect for scientific ethics of which we are very proud.

This article is 22 pages length, contains 13 Figures, 4 Tables, 1 scheme + supplementary information.

We have performed X-ray fluorescence analysis, nitrogen adsorption/desorption, X-ray analysis, Raman spectroscopy, transmission electron microscopy with analysis of the surface composition by X-ray energy dispersive spectroscopy and X-ray photo-electron spectroscopy. In our opinion, we have provided very satisfactory characterizations. We totally disagree on the sentence: “Publishing a paper without evidence and proper study on the catalyst’s characteristics is not ethical and not acceptable scientifically”!!

Any new investigation about the effect of Ce/Mn molar ratio will be addressed in a future work.
